# Investigation of Deep-Sea Ecosystems Using Marker Fatty Acids: Sources of Essential Polyunsaturated Fatty Acids in Abyssal Megafauna

**DOI:** 10.3390/md20010017

**Published:** 2021-12-23

**Authors:** Vasily I. Svetashev

**Affiliations:** A.V. Zhirmunsky National Scientific Center of Marine Biology, Far Eastern Branch, Russian Academy of Sciences, 17 Palchevskogo Str., Vladivostok 690041, Russia; vsvetashev@mail.ru

**Keywords:** PUFA, fatty acids, abyssal, deep-sea, food webs, megafauna, invertebrates, foraminifera, nematodes, bacteria

## Abstract

Abyssal seafloor ecosystems cover more than 50% of the Earth’s surface. Being formed by mainly heterotrophic organisms, they depend on the flux of particulate organic matter (POM) photosynthetically produced in the surface layer of the ocean. As dead phytoplankton sinks from the euphotic to the abyssal zone, the trophic value of POM and the concentration of essential polyunsaturated fatty acids (PUFA) decrease. This results in pronounced food periodicity and limitations for bottom dwellers. Deep-sea invertebrate seston eaters and surface deposit feeders consume the sinking POM. Other invertebrates utilize different food items that have undergone a trophic upgrade, with PUFA synthesized from saturated and monounsaturated FA. Foraminifera and nematodes can synthesize arachidonic acid (AA), eicosapentaenoic acid (EPA), while some barophylic bacteria produce EPA and/or docosahexaenoic acid. FA analysis of deep-sea invertebrates has shown high levels of PUFA including, in particular, arachidonic acid, bacterial FA, and a vast number of new and uncommon fatty acids such as 21:4(n-7), 22:4(n-8), 23:4(n-9), and 22:5(n-5) characteristic of foraminifera. We suppose that bacteria growing on detritus having a low trophic value provide the first trophic upgrading of organic matter for foraminifera and nematodes. In turn, these metazoans perform the second-stage upgrading for megafauna invertebrates. Deep-sea megafauna, including major members of Echinodermata, Mollusca, and Polychaeta display FA markers characteristic of bacteria, foraminifera, and nematodes and reveal new markers in the food chain.

## 1. Introduction: General Characteristics of the Abyssal Zone

Abyssal ecosystems located at depths of 3000–6000 m occupy almost 55% of the Earth’s surface. These ecosystems are characterized by relatively stable conditions: high hydrostatic pressure, water salinity, oxygen content, and low water temperature (0–3 °C) [1,2,3]. The abyssal seafloor is mostly covered by muddy sediments. In addition, ferromanganese nodules that are widespread in the deep-sea zones may cover more than 50% of the seafloor, especially in the Pacific Ocean [3,4]. Abyssal bottom ecosystems are characterized by a lack of in situ primary production (except those formed near hydrothermal vents and cold seeps). Only around 1% of surface photosynthesis products reach the bottom [5,6,7]. Particulate organic matter (POM), which is a major source of energy in the benthic environment, mainly consists of the remains of planktonic organisms, fecal pellets, and molts. The POM reaching the seafloor after being subject to complex biological modifications during sinking has a relatively low nutritional value [8,9,10]. This concerns, in particular, the level of essential polyunsaturated fatty acids (PUFA) which are necessary for the normal development of abyssal meio-, macro-, and megafauna [11,12,13]. Major essential PUFA for marine invertebrates are eicosapentaenoic (EPA), docosahexaenoic (DHA), and, to a lesser degree, arachidonic acids synthesized mainly by microalgae in the photosynthesis zone of the ocean [11]. These PUFA are crucially important for the normal functioning of biological membranes under extreme environmental conditions in the abyssal zone [14,15,16]. In this review, we consider the FA compositions of deep-sea sediments, detritus, bacteria, protists, and macro- and mega-fauna. Special attention is paid to marker fatty acids and trophic upgrading in abyssal ecosystems.

## 2. Some Notes on Samples Used for Lipid Extraction and the Methods of Fatty Acid Analysis

Deep-sea samples are not always available in sufficient amounts for dissection; some of them are subject to damage and loss of integrity during collection and retrieval to the surface. Small animals are difficult or impossible to dissect for the separation of tissues, so they are extracted whole. In the cited articles, for the analysis of holothurian lipids, the body wall was used; for sea stars, tube feet; and for sea urchins, total guts after washing out sediments. In mollusks, muscle tissue was analyzed. More detailed information about sample treatment can be found in the cited articles.

To date, a total of more than 1000 fatty acids have been found in nature, with this number constantly growing. Currently, analysis of fatty acids as methyl ester derivatives (FAME) is almost a routine procedure. Gas chromatography of a complex mixture of FAME on a carbowax-based (Supelcowax, DB-Wax, or similar) polar column takes less than 1 h. FAME peaks in a GC chromatogram are usually identified using various commercial FAME standards. The major problem is the lack of standards for all fatty acids. The second most efficient method for identification is the calculation of equivalent chain length (ECL) values by comparing them to published data [17]. Furthermore, one can compare the calculated ECL to theoretical values and even suggest the structure of the acid [18,19]. FA analysis of abyssal organisms showed many unusual peaks of FA in chromatograms (Figure 1), which are absent in the lipids of shallow-water marine invertebrates [20,21,22]. Some of them were present at a concentration of up to 6 mol%. For more accurate identification, two types of capillary columns should be used: polar columns, which provide a good separation of most FAME, and non-polar columns for the analysis of acids with carbon chains longer than 24 carbon atoms. For example, FAME 22:4(n-8), 22:5(n-5), and 22:6(n-3) form a single unresolved peak on an MDN-5S non-polar column, whereas these FAME are separated on a polar column. A major source of structural data is GC–MS of FAME, dimethyloxazoline (DMOX), and pyrrolidide derivatives. In general, the MS spectra of FAME give limited information: molecular mass and relation to (n-3) or (n-6) family. The spectra of DMOX derivatives are useful for determining the positions of all double bonds by GC–MS. An important advantage of these derivatives is the low temperature of separation, only by 5–10 °C higher than for FAME. It gives an opportunity to use them for GC–MS analysis of DMOX derivatives on a polar column. A mild method for the preparation of DMOX derivatives, which is performed at a low temperature, has been proposed recently [23]. Pyrrolidide derivatives of FA are more suitable for structural analysis of monoenoic and branched-chain FA [24]. Analysis of octadecapentaenoic acid 18:5(n-3) is also worth mentioning here. This acid is a characteristic component of lipids of many microalgae, as well as lipids of photosynthetic dinoflagellates. Its concentration varied from 3 to 43% of total lipids. However, the use of esterification methods based on the preliminary saponification of lipids or base-catalyzed transesterification resulted in a mixture of double-bond positional isomers of the 18:5 acid [25]. The best and simplest way to avoid incorrect GC results is using the standard acid catalyzed methylation.

## 3. Fatty Acids of Abyssal Bottom Sediments

Sediments are a complex mixture of components sinking from the ocean’s photosynthetic zone. Particulate organic carbon (POC) is transported to the bottom as detritus aggregates loaded with inorganic particles including mainly calcite CaCO_3_ and biogenic opal. Most POC is remineralized through metabolic processes in the epipelagic ecosystem. The remaining POC, constituting roughly one-fifth of the primary production (PP), leaves the epipelagic zone as the export production of carbon (EP) [6]. Eventually, only approximately 1% of initial PP reaches the seafloor [4]. Levels of major biological components—total organic carbon, nitrogen, and lipids—significantly decrease with depth [8,9,26,27]. The lipid content and fatty acid composition of POC and sediments vary widely. In the Western Crozet Basin (Indian Ocean, depths 3615–4750 m, four stations), the total lipid (TL) content was from 74 to 1033 μg/L in overlying particles and fluffs, and from 24 to 97 μg/g dry weight (DW) in surficial sediments [28]. The PUFA content was 1–3%, represented mainly by AA, EPA, and DHA. The major MUFA were 16:1(n-7), 18:1(n-7), and 18:1(n-9). Branched and cyclic FA made up 7–13%, and the total proportion of bacterial markers (branched, cyclic, 18:1(n-7), and SFA > C20) reached 33–46% [28]. Similarly, in sediments from the Cape Verde Plateau (depth 3100 m), the concentration of TL, mainly PL, was 0.22 mg/g DW; the PUFA (EPA, DHA, and AA) content was 7.2%; and bacterial FA content was 19% [29]. A significant PUFA (DHA, EPA, and AA) content, 4–10% in sediment and 5–14% in POM, and a relatively low bacterial FA content were recorded from deep-sea sediments collected in the Southern Ocean [30,31]. Nevertheless, reports about the lack of PUFA in sediments are quite usual [9,32]. In deep-sea sediments, the major components are generally saturated and monoenoic acids and a low level of PUFA. Due to the variation in data on lipids and fatty acids between different regions and stations, it would be better to consider the results of FA analysis for deep-sea megafauna along with information on reference sediments.

## 4. Fatty Acids of Foraminifera

Foraminifera are a major component of a deep-sea benthic community [33,34,35]. They are ubiquitously distributed in marine sediments, from shallow to deep-sea waters of the ocean. Foraminifera’s significance and relative biomass increase with depth, reaching 50% of top sediment biomass [34]. The comparable biomasses of foraminifera and metazoan meiofauna suggest that both groups may play an important role in functions of the deep-sea benthic community and, therefore, in carbon cycling [33]. According to [36], foraminifera are probably the most important eukaryotic members of abyssal sediment communities. The trophic diversity of foraminifera is well known. Bathyal species include herbivores and opportunistic deposit feeders (omnivores) that consume labile organic matter. Abyssal foraminifera ingests sediment, associated bacteria, and more refractory organic matter [36,37], while some species can be carnivorous [38]. Deep-sea foraminifera species feed on bacteria by ingesting bacterial cells, in addition to relatively large volumes of associated sediment and organic detritus, during deposit-feeding [39]. It can be assumed that the trophic role of labile organic matter decreases with depth, and the major food supply for foraminifera in the abyssal zone are low-value detritus and sediment bacteria. Accordingly, benthic foraminifera can be a trophic link between low-value phytodetritus and bacteria and benthic metazoans [40,41]. Since the input of detritus depends on the place and time of sample collection, it would be more correct to analyze foraminifera and reference sediments in parallel, where possible.

Deep-sea abyssal foraminifera generally show some specific FA composition. In the deep-sea *Bathysiphon capillary*, MUFA (18:1(n-7), 20:1(n-9), and 22:1(n-7)) were found at a level of 39% [42]. The AA content exceeded the level of 20:5(n-3) and 22:6(n-3). Moreover, C20 and C22 non-methylene interrupted (NMID) acids of bacterial origin were found in significant amounts [43,44]. In phospholipids of the deep-sea *Xenophyophore*, an exceptionally high level of bacterial FA (33%) and small amounts of AA, EPA, and DHA (8.4% in total) were recorded [29]. In lipids of reference sediments, the bacterial FA content was 18.8%, and the total PUFA content was 6.4% [31]. Mixed samples of foraminifera from the Antarctic abyssal showed noticeable amounts of AA, EPA, DHA, and 18:3(n-6) acids. The level of odd- and branched-chain (bacterial) FA was low, ~3%. Furthermore, the levels of these FA in reference POM and sediments were <1%. There was also a noticeable ratio, 18:1(n-9)/18:1(n-7) > 1, characteristic of phytoplankton. Experiments in situ on a lander system at a depth of 140 m and in an on-board laboratory showed a remarkable accumulation of arachidonic acid (10-fold) and 18:1(n-7) in the foraminifera *Uvigerina* ex. Gr. *Semiornata* after being incubated with a ^13^C-labeled diatom, *Thalassiosira weisflogii.* [45]. The authors suggested this foraminifera to carry out its own biosynthesis of arachidonic acid and discussed its role as a possible major source of AA in the benthic food web. Similar results and conclusions on AA biosynthesis were also reported for the benthic shallow-water foraminifera *Ammonia tepida* [46]. In the common benthic foraminifera *A. tepida* feeding on diatoms in oxic and anoxic conditions, the major PUFA were EPA > AA > DHA > 22:5(n-3). However, only arachidonic acid demonstrated the most significant increase. In the Antarctic shelf (depth 570 m), three species of benthic foraminifera showed selective feeding on phytodetritus [47,48]. The authors found a low PUFA content ranging from 5.4 to 30% and, accordingly, low levels of AA (0.8–9.4%), EPA (0.2–9.4%), and DHA (1.1–5.1%). The level of bacterial FA varied from 5 to 11.4%. In the Southern Ocean, a deep-sea mixed sample of foraminifera was found to contain a high proportion of 20:4(n-6), up to 21% of total FA, but not in reference sediments and POM [31]. The level of EPA and DHA was lower than AA. Oleic (18:1(n-9)) and *cis*-vaccenic 18:1(n-11) acids were at comparable concentrations. An analysis of the FA of four foraminifera species (*Bathysiphon lanosum*, *B. major*, *Rhabdamina abyssorum*, and *Rh*. *parabyssorum*) from the abyssal zone off the Kuril Islands showed a very high PUFA content, 46.8–53.8%, while the MUFA and SFA contents were 28–43% and 9.6–25%, respectively [20]. Arachidonic acid was a major PUFA, reaching up to 28%, and *cis-vaccenic* acid was a major MUFA reaching 25.3%. Representatives of the two genera were similar in PUFA content but showed significant variations in AA, EPA, and DHA contents. Moreover, these species had high levels of branched and odd-chain acids, 14 and 5.5%, respectively. With one more bacterial FA 18:1(n-7) taken into account, the total of bacterial FA reached 30%. *B. lanosum* and *B. major* had remarkable levels of NMID acids, 20.8 and 10.7%, respectively [20]. These foraminifera species were found to contain 10 new, and a number of uncommon fatty acids, mainly monoenoic and dienoic with the first double bond at positions Δ4 and Δ7 (Table 1). The levels of the new acids were quite high: 20:2(n-13) amounted to 8.8%; 22:2Δ7,12, 22:4(n-8), and 23:4(n-9) amounted to 2%. It is worth mentioning that the latter two acids are members of the homologous series of arachidonic acid, 20:4(n-6), 21:4(n-7), 22:4(n-8), and 23:4(n-9). The first acid in this series, 21:4(n-7), was first found in the amphipod *Pontoporeia femorata* [48] and later in thraustochytrids [49]. The authors suggested a way of biosynthesis which is also suitable for 22:4(n-8) and 23:4(n-9) acids [50]. In a recently published paper, rare and uncommon PUFA in abyssal foraminifera have been considered [21]. Lipids of *Reophax nodulosus* were found to contain 8% odd- and branched-chain FA, 17% SFA, 22% MUFA, and 53% PUFA. The AA content was two-fold higher than EPA; DHA constituted only 1%. *Cis*-vaccenic acid was the major MUFA. In addition, new acids of the omega-5 family were revealed: 20:3(n-5), 22:3(n-5), and 22:5(n-5). The latter of them was a major component at a concentration of 18.3% [21]. Earlier, omega-5 FA with odd carbon chains was also found in thraustochytrids [49]. One species, *Pyrgo* sp. Showed the presence of one more homologous series of FA related to 22:4(n-6): 23:4(n-7), 24:4(n-8), and 25:4(n-9). The author suggested a possible pathway of biosynthesis [21].

Indeed, the FA composition of foraminifera depends on available food. When available food is fresh detritus, foraminifera’s FA consist of mainly phytoplankton acids. With other diets (recalcitrant organic matter, bacterial, omnivorous, or carnivorous), FA composition can be quite different. Nevertheless, it is possible to find common features in foraminifera from different biotopes. Deep-sea foraminifera are characterized by high levels of PUFA, ~50% of total FA, with a major component (up to 25%) being arachidonic acid, which exceeds the EPA and DHA contents [20,31]. Foraminifera are capable of synthesizing arachidonic acid. Thus, they are possibly a major source of this acid in deep-sea ecosystems. The major monoenoic acids are 18:1(n-7) which originate from both detritus and bacteria and 18:1(n-9) at a ratio of > 1. Another characteristic feature is an exceptionally high concentration of bacterial branched and odd-chain FA (reaching in some cases 33%) and a high level (up to 20%) of non-methylene interrupted dienoic FA, which is also initially of bacterial origin. Some abyssal foraminifera species contain a number of uncommon and new FA, which can be a product of their own biosynthesis or modification of FA from bacteria ingested as food or biosynthesis in symbionts closely associated with these foraminifera. Table 1.

## 5. Fatty Acids of Nematodes

Nematodes are the most diverse, abundant, and very important animals in deep-sea sediments. They represent a very successful higher metazoan taxon, making up from 90 to 99% of the total metazoan abundance [51,52]. Nematodes become increasingly dominant, in terms of relative abundance and species richness, with depth. Free-living aquatic nematodes may feed on a diverse array of resources such as bacteria, Achaeans, protists, fungi, particulate and dissolved organic matter, and as predators prey on metazoans, including other nematodes [53]. Some free-living deep-sea nematode species are associated with symbiotic bacteria [54]. Nematodes may be an important source of PUFA for larger animals in deep-sea environments, where the nutritional quality of sediment organic matter is low [32].

One of the first reports on PUFA biosynthesis in the free-living nematode *Turbatrix aceti* was the article published in 1968 [55]. The authors clearly showed the synthesis of AA, 20:3(n-6), and EPA. The nematodes *Caenorhabditis elegans* and *C*. *briggsae*, when cultivated on complex media, also synthesized PUFA 18:2, 20:2 20:3, and 20:4, all of the (n-6) series [56]. Only three ω3 acids, EPA, 20:4(n-3), and 18:3(n-3), were found. DHA was absent, and the amount of 18:1(n-7) prevailed over more common 18:1(n-9) [56]. The nematode *Panagrellus redivivus* growing on different media showed quite similar FA compositions: the lack of DHA, AA, and 20:3(n-6) > EPA, and a high ratio of (n-6)/(n-3) 5–18, 18:1(n-7) > (n-9), and 18:0 > 16:0. The presence of different desaturases and elongases suitable for synthesizing (n-3) PUFA was also shown [57]. These results were consistent with earlier data by [58]. Similar results were obtained in experiments with nematodes that had been fed *E. coli*. The nematode *C. elegans* accumulated mainly 18:1(n-7) (16%) and only a small amount of 18:1(n-9). The major PUFA were EPA, AA, 18:2(n-6), 20:3(n-6), and 20:4(n-3). The level of bacterial FA, including cyclopropanoic acid, was almost 12% [59]. The results obtained by [60] in a study of FA from mutant *C. elegans* deficient in PUFA synthesis showed that control wild-type *C. elegans* contained 27% bacterial FA, 21% 18:1(n-7), 1.7% AA, and 19.1% EPA.

In a shallow-water nematode, bacterial FA varied in a range of 6–12%, PUFA in a range of 32–47%, and the DHA/EPA and 16:0/16:1n7 ratio was >1. Such composition suggests a flagellate-based diet. In reference sediments, the PUFA content was five-fold lower [61]. Experiments on the effect of deep-sea bacterivorous nematodes on detritus [62] showed the presence of 18:1(n-7) > (n-9), 18:0 > 16:0, EPA >> AA, and the absence of DHA in *Rhabditis mediterranea*. In the deep-sea nematode *Deontostoma tridentum* from the Chatham Rise, the major unsaturated FA were 16:1(n-7), 18:1(n-9) >> 18:1(n-7), DHA > EPA > AA, and the PUFA content was only ~23%. Moreover, no PUFA were detected in reference sediments [32]. Mixed samples of nematodes from deep-sea sediments of the Southern Ocean [63] showed a low lipid content because these nematodes do not accumulate lipids for energy storage and may feed throughout the year on constantly available food sources. The total PUFA was 33–45% and consisted mainly of FA of planktonic origin. The levels of major PUFA were markedly high (AA, 6–14%; EPA, 7–12%; DHA, 10–20%), while the amount of bacterial odd- and branched-chain FA was low [63]. Abyssal nematodes *Desmodora* and members of Desmoscolecids from the Southern Ocean were assumed to have a plankton-based diet, as evidenced by diatom biomarkers 16:1(n-7)/16:0 > 1. Furthermore, AA, known as a common FA in foraminifera, was found in high abundances (12.7–23%) in all nematodes. The level of bacterial FA was up to 6.1%. The DHA values reached almost 20% of total FA. It is known that nematodes are capable of producing n-3 and n-6 PUFA, but not DHA, which indicates a selective feeding behavior [64]. The FA composition in the deep-sea nematode *Halomonhystera hermesi* from the Håkon Mosby mud volcano showed a low level of SFA (9%), monoenoic FA (54%) represented mainly by 16:1(n-7) and 18:1(n-9), and PUFA (37%), mostly 18:2(n-6), DHA, EPA, and AA. It is known that the food sources of *H*. *hermesi* lack EPA and DHA, which suggests that this nematode most likely exhibits the ability to synthesize these PUFA [65]. Four nematode species from the abyssal zone in the Kuril Basin (3300–4700 m) demonstrated a high level of bacterial FA (13–22%), phytoplankton FA (31–34%), and monoenoic acids 20:1 and 22:1 (19%). With an odd and branched chain with 18:1(n-7) taken into account, the total bacterial FA content can reach 42–64%. Organic matter derived from chemosynthetic bacteria is one of the carbon sources for macrobenthic nematodes. The monoenoic FA were 18:1(n-9), (5–28%), and 18:1(n-7) (5–13%). The major PUFA were AA (4–13%) and EPA (4–17%); the level of DHA was 1–12% [66].

In conclusion, experiments on free-living nematodes with the use of ^13^C labeled precursors and a diet with specific FA composition have demonstrated the biosynthesis of C18 and C20 PUFA. A ratio of 18:0 > 16:0 and 18:1(n-7) > 18:1(n-9) independent of diet is characteristic of nematodes. It is also reported for nematodes growth on the E. coli diet. Thus, a ratio of 18:0/16:0 and 18:1(n-7)/18:1(n-9) > 1 can be a characteristic marker of nematodes. For PUFA, the most noticeable characteristics are the lack of docosahexaenoic acid 22:6(n-3), high levels of both AA and EPA, and the presence of different C18, C20, n-3, and n-6 precursors of these acids. Therefore, and vice versa, the occurrence of DHA and 18:1(n-9) in deep-sea nematodes in significant amounts can be linked with the specific predation on meiobenthos. For bacterivorous nematodes, a characteristic feature can be the accumulation of bacterial FA to 10% and more. There is data on the occurrence of desaturase and elongase activities in nematodes leading to different intermediate unsaturated C18 and C20 acids and to eicosapentaenoic acid as a final product [60,67,68].

## 6. Abyssal Megafauna

### 6.1. Holothurians

The results of studies on the distribution of invertebrates in the abyssal zone of the Pacific, Indian, and Atlantic oceans are presented and compared in an extensive review considering the trophic structure of deep-sea macrobenthos (with sizes larger than 5 mm) [69]. A total of 118 benthic species were analyzed, of which most were Echinodermata (78 species, 66%) represented mainly by Holothuroidea, Ophiuroidea, and Asteroidea. Holothurians made up about 76% and 93% of the megabenthos in terms of abundance and biomass, respectively [70]. In favorable conditions, their biomasses can be exceptionally high, which increases in response to the organic matter flux [70]. Holothurians dominate soft sediments; moreover, they are major phytodetritus consumers and amend reworked sediments for other organisms [71]. There is also data showing that they feed on bacteria or a mixed diet including bacteria and detritus [72,73]. The first report on the FA composition of the abyssal (4400 m) holothurian *Scotoplanes theeli* was published in 1967 [74]. It stated that the presence of arachidonic acid reaching 21% in TL of the holothurian “has no ready explanation”. Moreover, a remarkable level of odd- and branched-chain acids and a small concentration of DHA were recorded. In East Atlantic abyssal holothurians [75], FAs were identified not fully, but nevertheless, sufficient to show very high concentrations of 23:1 and 24:1 acids (up to 17% and 16%, respectively). AA and EPA amounted to about 15%; bacterial FA, 8%; 20:1 and 22:1 acids, up to 15%. However, 18:1 acids were not identified. The presence of long-chain monoenoic acids 20:1 and 22:1 was explained by feeding on zooplankton remains. An extensive study of deep-sea holothurian lipids has been carried out [76]. The FA composition was found to be subject to significant seasonal variations. In all seasons, lipids were dominated by PUFA, about 50–60%; AA, EPA, and DHA varied widely (12–29%, 6–27%, and 3–12%, respectively); the ratio of 18:1(n-7)/18:1(n-9) was > 1. Odd- and branched-chain FA and 23:1 were not found. At the same time, non-methylene interrupted dienoic acids (NMID), which are also of bacterial origin, were detected at high levels [44]. The authors explained the variations in FA by seasonal detritus input and reproduction. Four species of East Pacific abyssal holothurians showed the presence of PUFA (29–45%), MUFA (28–45%), bacterial FA (3–12%), and DHA (1–8%). In all species, the major PUFA were EPA > AA > DHA. The authors found also uncommon acids 2OH-23:1, 2OH-24:1, and 23:1. This FA composition suggests mixed diets consisting of detritus, bacteria, and zooplankton remains [77]. Holothurians from the West Pacific abyssal plain also showed similar levels of EPA > AA > DHA, odd- and branched-chain FA (3.4–13.9%), and C20 and C22 monoenoic FA. It means a significant input of fresh detritus, bacteria, and zooplankton remains [71]. Of particular note is the presence of an unusual acid, 23:1(n-9), which previously was discovered in shallow-water holothurians [78]. In bathyal holothurians from the Sea of Okhotsk (depth 90–560 m), FA from three groups with different nutrition modes were found: PUFA, up to 64%; MUFA, 20–33%; and bacterial FA, 3.5–6.9% [79]. The first group of suspension feeders had the highest concentration of EPA, 38.8%; the second group of surface feeders had high levels of both AA and DHA; the third group of surface and subsurface feeders was distinguished by the highest levels of AA (up to 39%) and uncommon acids 21:4(n-6) and 23:1(n-9) [79]. Later [80], the holothurian *Molpadia musculus*, a widespread subsurface deposit feeder, was studied at five Kuril Basin abyssal stations at a depth of 3500 m. The total PLFA was in a range of 34–40%; AA dominated at all stations (up to 26%); EPA and DHA were at a much lower concentration, only ~8 and 1%. The bacterial FA content was almost 10%, with a ratio of 18:1(n-7)/18:1(n-9) > 1. The total of C20:1 and 22:1 was about 5–7%; the levels of uncommon FA 23:1(n-9) and 21:4(n-7) were 7–9% and 1.1–1.3%, respectively. In general, the authors found an insignificant difference in FA composition between stations. Other surface-feeder holothurians from the same station, *Psychropotes raripes* and *Peniagone dubia* had much higher values of EPA and DHA concentrations and an EPA/AA ratio. The authors suggested using the ratio EPA/AA and DHA/EPA as an index of the contribution of foraminifera to the diet of deposit feeder [80].

### 6.2. Sea Stars, Brittle Stars, and Sea Urchins

Sea stars are the second-highest in abundance and importance group of Echinodermata after holothurians in deep-sea bottom habitats [70,81]. Nine species of sea stars from the abyssal zone of the NE Atlantic Ocean were investigated for lipids and FA and divided into three trophic groups on the basis of FA markers [81]. The total PUFA content in all groups was generally high (40–47%); the major FA were EPA and AA, and in all samples, the ratio 18:1(n-9)/18:1(n-7) was <1. In suspension feeders, the main components were EPA and DHA, characteristic of photosynthetic microplankton; the ratio EPA/AA was 1.7–2.6. Moreover, this group had the highest level of C20:1 and C22:1 acids. In the group of mud ingesters, the main PUFA were AA and EPA, but the EPA/AA ratio was 0.5–0.7, and the total level of bacterial FA, 18:1(n-7), and NMID acids amounted to 7.7–9.5%. The amount of odd- and branched-chain FA was low, 1.1–3%. The group of predator/scavengers had an EPA/AA ratio of 0.8–1.1%. An analysis of FA of the sea star *Eremicaster vicinus* from the Kuril–Kamchatka Trench (5210 m) showed a PUFA content of 37.8%; MUFA, 42.2%; and odd- and branched-chain acids, 17.35%. The EPA/AA ratio was 1.0; the 18:1(n-9)/18:1(n-7) ratio, 0.1. Such ratios and the very high concentration of odd-branched chain FA can indicate that *E*. *vicinus* feeds on foraminifera, which in turn feeds on bacteria. In addition to common fatty acids, a number of new and uncommon FA: 21:4(n-7), 22:4(n-8), 22:5(n-5), and 23:4(n-9) earlier discovered in deep-sea foraminifera were also found (Table 1) [20]. Furthermore, a new acid Δ5,8,11,14,17,20, 22:6 or 22:6(n-2), related to the ω2 family, was also detected. Earlier acids 21:5(n-2) and 17:5(n-2) were tentatively identified in lipids of the amphipod *Pontoporeia femorata*, which had a remarkable level (up to 53%) of odd-chain FA [48]. The acid 21:6(n-2), another member of the ω2 family, was detected in thraustochytrids [51]. Unusual FA 21:4(n-7) 5% and minor 22:4(n-8) and 22:5(n-5) were found in abyssal Asteroidea [21].

Information on FA of deep-sea ophiuroids is provided only in a few papers [77]. Two species of brittle stars from the abyssal zone of the Northeast Pacific had a high EPA/AA ratio (3.4–4.2) and a total level of C20 monoenoic FA of 17–19%. Ophiuroids are known to be opportunistic feeders consuming both phytodetritus and animal-derived matter, including foraminifera and nematodes [82]. However, the high level of C20 monoenoic acids suggests feeding on zooplankton remains. Deep-sea ophiurans showed the presence of an uncommon very long-chain PUFA 24:6(n-3) (2.3–10.3%) and an uncharacterized C26 PUFA (10.6–0.7%) [77]. Subsequently, these acids were found in bathyal brittle stars and were characterized as a family of C26 PUFA: 26:7(n-3), 26:6(n-3), 26:6(n-6), and 26:5(n-3) [83]. Moreover, high levels of 24:6(n-3) and 26:6(n-3) PUFA (constituting 20.5% and 14.5% of total FA) were found in brittle stars from the abyssal zone off the Kuril Islands [21]. An analysis of one brittle star species (among many other invertebrates), *Ophiura leptostenia*, from the Sea of Japan [84] showed the results generally consistent with data of [77]. The EPA/AA ratio was high, 11.4 and 16.4% at two stations; the ratio 18:1(n-9)/18:1(n-7) was 1.6 and 2.1, respectively. The total of C20 MUFA was 10% at both stations. The concentration of a “marker” FA for brittle stars, 24:6(n-3), was 13.7 and 4.9%, respectively. The second marker FA 26:7(n-3) probably got into the proportion of “other”, 8.5%. The low level of bacterial acids, 18:1(n-7), and the high EPA/AA and EPA/DHA ratios indicate that brittle stars feed mainly on detritus and zooplankton [84]. In reference sediments (depth 2500 m), noticeable levels of PUFA 18:2(n-6) (5–11%), EPA (0.9–6%), AA (0.3–2%), DHA (0.3–0.9%), and 18:1(n-9) as a major MUFA (17%) were found. The concentration of odd- and branched-chain FA was 7.9%. The presence of C18, C20, and C22 PUFA (up to 18%) in significant amounts in sediments allows an assumption about their important role in the diet of surface feeders [84].

The prevalence of eicosapentaenoic acid as major PUFA, characteristic for fresh detritus, over arachidonic acid, 18:1(n-7) vs. 18:1(n-9), the low concentration of bacterial odd- and branched-chain FA, and the high level of C20 monoenoic acids indicate a mixed detritus/-zooplankton diet. Uncommon 24:6(n-3) and 26:7(n-3) acids can be marker FA for deep-sea brittle stars.

Lipids and FA of shallow-water sea urchins are well known. For example [85], TL and FA data were obtained for the common sea urchin *Strongylocentrotus droebachiensis* fed different diets. Most PUFA were C18 and C20 acids derived from macroalgae. The FA found only in sea urchins (absent from their diet) were 20:1n-9, 20:1n-7, 22:1n-9, and several NMID (20:2Δ5,11; 20:2Δ5,13; 20:2Δ5,11; and 20:2Δ5,13). Data on FA of deep-sea sea urchins is limited. An analysis of FA of *Kamptosoma abyssale* from the Kuril–Kamchatka Trench (5210 m) showed a PUFA level of 44.4%, equal concentrations of EPA and AA (17%), and a small level of DHA [22]. The total of isomers C20:1 was 22%, with the major acid being 20:1(n-5), and the NMID acids content being low. The presence of uncommon FA characteristic of foraminifera—21:4(n-7), 22:4(n-8), 23:4(n-9), and 22:5(n-5)—found in the sample, allows an assumption that *K*. *abyssale* feed on foraminifera [22]. An investigation of FA of sea urchins (*Echinus affinus*) collected from a depth of 2700 m off the deep-sea dumping site located 185 km southeast of the New Jersey coast showed a high level of PUFA of about 70% with equivalent EPA and AA contents [86]. The level of DHA was low (2.7%), as well as the level of bacterial odd- and branched-chain FA (3.4%) with NMID acids, which are also of bacterial origin, with the total level being 7.6%. The 18:1(n-9)/18:1(n-7) ratio was almost 1. The authors explained the relatively high concentration of AA by possible biosynthesis from 18:2(n-6).

### 6.3. Mollusks

A comparative study of 12 species of abyssal mollusks (gastropods, bivalves, and scaphopods) from the Kuril–Kamchatka Trench was carried out by [87]. First, they showed that the FA composition in most of them depends on feeding type, rather than on taxonomic classification. On the basis of the FA profile, the studied mollusks were clustered into three groups. Group 1 included species with a high level of AA and bacterial FA including 18:1(n-7). The combination of FA found in this group is typical of organisms feeding on foraminifera [31]. Group 2 mollusks feed mainly on sediment surface layers and, therefore, they contained the highest amount of DHA, while EPA/AA was >1. DHA is highly conserved through the food chain and often increases towards higher trophic levels [88]. This could be indicative of carnivorous feeding. Group 3 contained a high level of EPA, 22:5(n-3) and 20:1(n-7). Two gastropod species are known to exhibit carnivorous/scavenging feeding strategies. The significant accumulation of 22:5(n-3) cannot be easily explained [87]. The detection of up to 10% of uncommon acid 21:4(n-7), probably of microbial origin in most mollusks studied deserves special mention [89]. An FA analysis of particulate organic matter (POM) from the reference bottom-water interface revealed a significant amount of 16:1(n-7), 18:1(n-9), 18:1(n-7), whereas no appreciable amount of PUFA, except 18:2(n-6), were present in the samples [87].

### 6.4. Polychaetes

In lipids of three polychaete species from the abyssal zone of the northeastern Pacific Ocean, the levels of PUFA were 35–45%, with major components being EPA, AA, and C22 PUFA [90]. Bacterial odd- and branched-chain FA contents were 5–8%, but the levels of NMID, also of bacterial origin, were 8–13%. Thus, the total content of bacterial acids was very high, up to 20%. C20 and C22 MUFA were also present at a noticeable level, 13–19%. The authors assumed the consumption of phytodetritus from the sediment surface and zooplankton-derived matter [90]. Extensive studies of 39 polychaete species from 18 families and reference sediments were performed by [30] on deep-sea samples from the Southern Ocean. In POM and sediment samples, the major FA were monoenoic and saturated. The levels of major PUFA, DHA, EPA, AA, and docosapentaenoic 22:5(n-3), varying from 1 to 4% in POM and 1–3% in sediments. In polychaetes, the major FA were monoenoic 18:1(n-7), 18:1(n-9), and 20:1(n-11) and PUFA 20:5(n-3), 22:6(n-3), 22:5(n-3), and 20:4(n-6). Polychaetes’ food sources such as fresh diatom remains, foraminifera, and zooplankton were identified using marker acids. Four polychaete species from the abyssal zone of the Kuril Basin, Sea of Okhotsk, were divided into two groups on the basis of 18:1(n-9)/18:1(n-7), EPA/AA, and DHA/EPA ratios [80]. Surface deposit feeders had an EPA/AA ratio of 0.7 and an 18:1(n-9)/18:1(n-7) ratio of 0.2. In the second group of surface deposit feeders, feeding in the most superficial layer of bottom sediments, these ratios were 2 and 1, respectively. The first group, besides higher levels of AA and 18:1(n-7), had two rare acids, 22:4(n-8) and 22:5(n-5), which had earlier been discovered in agglutinated foraminifera from the Kuril Basin [20]. These data point to foraminifera’s contribution to the diet of polychaetes. The second group had the FA characteristic of fresh detritus in the diet.

## 7. Discussion: Sources of PUFA in Abyssal Ecosystems

In deep-sea ecosystems exposed to a constantly low temperature and extremely high hydrostatic pressure, very high levels of essential PUFA (EPA, DHA, and AA) are required for providing normal functions of cell membranes [14,15,91,92,93]. It is generally recognized that abyssal ecosystems are fueled by the flux of POM from the ocean’s photosynthesis zone. As the organic matter sinks from the euphotic to the abyssal zone, the percentage of PUFA is reduced, and the trophic value of organic matter decreases [8,9,26,27,28,32,94,95,96]. However, there are reports about noticeable, up to 4–10%, concentrations of PUFA in sediments of the Southern Ocean, Kuril Basin, and Venezuela Basin deep-sea stations [29,30,31,50,93]. Such data point out that detritus becomes a source of essential PUFA for the megafauna, at least periodically. As a rule, high levels of EPA and DHA in deep-sea invertebrates are associated with detritus containing marked amounts of these acids.

Bacteria, which are actually present at all depths and constitute the major part of benthic biomass (11–87%), can be another possible source of essential PUFA [26]. Barophilic and psychrophilic bacteria are capable of producing EPA or DHA, while some strains produce both these acids, but not arachidonic acid [91,92,93]. It is difficult to access the input of barophilic bacteria to the sediment PUFA pool. All data on the FA composition of barophilic bacteria were obtained from strains grown under optimum laboratory conditions. According to most data, the PUFA content of phospholipids in sediments is low or absent. If PUFA are present, arachidonic acid, being not essential for barophilic bacteria, is also present, as well as EPA and DHA [28,29,30,31,97]. It is likely that psychrophilic and barophilic bacteria produce these PUFA in controlled laboratory conditions that are much more favorable than in natural habitats. Thus, to assess the input of bacterial PUFA to abyssal ecosystems, additional studies are needed.

The key role of foraminifera in marine food chains, as organisms recovering energy from low-value organic debris and bacteria, was formulated in 1970 [98]. Deep-sea foraminifera ingest planktonic and other detritus, sediment particles, bacteria, and more refractory organic matter, whereas some species can be specialized predators [4,5,6,7,34,38]. In turn, foraminifera are utilized as food by larger organisms and, therefore, they support the trophic upgrade of the ecosystem. Foraminifera constitute the major part of the deep-sea benthic community with relative biomass in the top layer of sediments reaching nearly 50% [2] and are comparable in biomass to the metazoan meiofauna [33,34]. It means that foraminifera plays a significant role in deep-sea food webs and carbon cycles. The ability of marine protists to synthesize PUFA is commonly recognized [99,100,101]. The most significant feature of foraminifera is the capability of synthesizing arachidonic acid under incubation with a ^13^C labeled diatom, *Thalassiosira weisflogii* [45]. A remarkable (10-fold) increase in the AA content and a less pronounced increase in 18:1(n-7) were recorded. Similar, but less noticeable results were reported for the common benthic foraminifera *Ammonia tepida* fed a diatom diet [46]. Both studies made conclusions that foraminifera can synthesize arachidonic acid and are possibly the major source of this acid in bottom ecosystems. An analysis of FA from abyssal foraminifera showed their dependence on a food source, a high PUFA content (up to 54%), and arachidonic acid as a major component, up to 28%. Moreover, a characteristic feature was the presence of odd- and branched-chain acids, NMID, and cis-vaccenic acid, all of bacterial origin. Thus, arachidonic acid is synthesized by foraminifera and then can be retained in the food chain. A detailed FA analysis using GC and GC–MS made it possible to detect 10 new acids in foraminifera, essentially monoenoic and dienoic ones with a first double bond at Δ4 and Δ7 positions (Table 1). Furthermore, two new PUFA were found, identified as 22:4(n-8) and 24:4(n-9), which form a homological sequence with arachidonic acids: 21:4(n-7), 22:4(n-8), and 23:4(n-9) [20]. Subsequently, two more homological series were identified: 20:3(n-5), 22:3(n-5), and 22:5(n-5); 22:4(n-6): 23:4(n-7), 24:4(n-8), and 25:4(n-9) [21]. Some of these acids were detected in members of megafauna dwelling in the surface and subsurface layers of abyssal sediments.

Another source of PUFA in deep-sea habitats is nematodes. Members of this phylum inhabit a broad range of environments and feed on diverse foods including bacteria, microalgae, fungi, small animals, fecal pellets, and dead organisms. Nematodes are usually small-sized organisms, for which it is difficult to obtain biomass sufficient for FA composition analysis on a species/genus level. Free-living marine nematodes are important and abundant members of the meiobenthos. Some experts estimate the total number of nematode species at around ~1 million [102]. The most important characteristic feature of nematodes is the de novo biosynthesis of a variety of PUFA from saturated and monoenoic acids [55,56,57,58,59,60,67,68]. It is worth noting that in *Caenorhabditis elegans*, grown on a pure culture of the bacterium *E. coli*, the major PUFA were EPA, AA, 18:2(n-6), 20:3(n-6), and 20:4(n-3), but not docosahexaenoic acid [59]. In contrast, the abyssal nematode *Halomonhystera hermesi* from the Håkon Mosby mud volcano, feeding on diets without EPA and DHA, nevertheless, accumulate 37% PUFA, mainly 18:2(n-6), DHA, EPA, and AA [65]. It means that nematodes are capable of producing DHA as well. A comparison between the FA compositions of nematodes from different stations has demonstrated the presence of FA characteristic of phytoplankton and zooplankton detritus, bacteria, and also FA characteristic of predators preying on meiobenthos. It can be assumed that nematodes, depending on available food supply, can extract PUFA from phyto- and zooplankton detritus or synthesize necessary PUFA from more available saturated and monoenoic acids, thus, upgrading the PUFA pool in sediments. Since the life span of free-living adult *Rhabdias bufonis* is only several days [103], they can significantly improve the nutritional value of sediments for various sediment-dwellers through bioturbation and PUFA biosynthesis.

## 8. Conclusions

In deep-sea ecosystems, there exist two major routes of PUFA entry: detritus flux from the photosynthesis zone and heterotrophic production by metazoans and, possibly, prokaryotes. The most important source of PUFA production is heterotrophic Protista which have a high biomass in deep-sea habitats and are capable of de novo synthesis of essential FA from low-quality detritus and bacteria [33,34,35]. Major representatives of Protista in the abyssal zone are foraminifera. The PUFA compositions of foraminifera from different places are always dominated by arachidonic acid. However, the level of odd- and branched-chain acids and 18:1(n-7) characteristic of bacteria can reach 33% in foraminifer’s phospholipids. One more consequence of feeding on bacteria can be the synthesis of new and uncommon fatty acids which can be markers of foraminifera in the further food chain.

The involvement of nematodes in the deep-sea PUFA pool is even more difficult to assess because they have no specific marker acids. However, nematodes were capable of synthesizing PUFA on bacterial feed in laboratory experiments, as well as in deep-sea sediments without any sources of PUFA. Having a short life span and forming significant biomass, nematodes undoubtedly improve the nutritional value of sediments for meiobenthos.

All studies on FA compositions of the major groups of deep-sea megafauna such as Echinodermata (holothurians, sea stars, sea urchins, and brittle stars), Mollusca (gastropods, bivalves, and scaphopods), and Polychaeta showed high levels of PUFA (40–50%). It seems that their FA compositions depend rather on the type of food supply than on taxonomic position. Nevertheless, some taxa have unusual acids or concentrations of common FA. Concentrations of AA, EPA, DHA, and some other polyenoic acids vary in a wide range. Based on concentrations of different PUFA or ratios of them, and also on the presence of FA characteristics for bacteria or zooplankton, it is possible to judge what type of food supply is essential for certain taxa.

## Figures and Tables

**Figure 1 marinedrugs-20-00017-f001:**
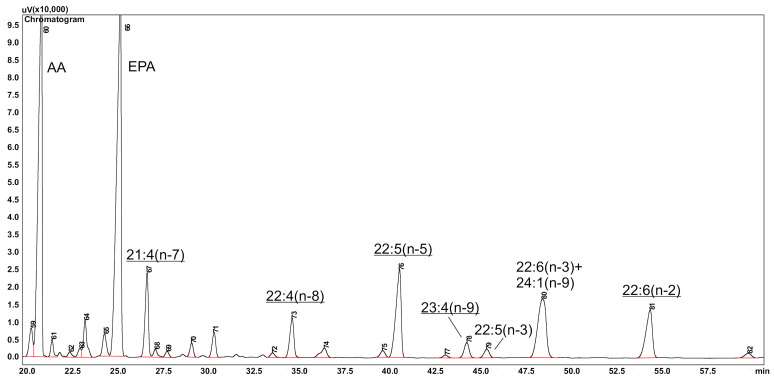
A part of the GC chromatogram of FAME from TL of the abyssal sea star *Eremicaster vicinus.* Conditions: Supelcowax 10 column, 30 m, at 205 °C, detector FID. FAME of 21:4(n-7), 22:4(n-8), 22:5(n-5), 23:4(n-9), and 22:6(n-2) acids had ECL values of 22.09, 22.98, 23.53, 23.82, and 24.53, respectively. Uncommon and new acids are underlined.

**Table 1 marinedrugs-20-00017-t001:** New and uncommon FA found in lipids of abyssal Foraminifera and Echinodermata. Data are presented as mol %; new acids are highlighted in bold. High concentrations are underlined.

FA	%	Species	Place and Depth	References
Foraminifera				
21:4(n-7)	2.4–4.8%	*Bathysiphon lanosum,* *B. major* *Rhabdammina abyssorum,* *Rh. Parabyssorum*	Kuril Basin, Sea of Okhotsk, 3307–3386 m	[20]
22:4(n-8)	1.2–2.0%			
23:4(n-9)	1.7–2.3%			
∆4-i-16:1	0.6–1.1%			
∆7-i-21:1	0.4%			
∆7-ai-21:1	0.3–0.8%			
∆4,11–18:2	0.3–0.9%			
∆4,7–20:2				
∆7,12–20:2	0.4–0.6%			
∆4,7–21:2	0.6%			
∆7,12–22:2	6.4–2.1%			
20:3(n-5)	1.2%	*Reophax nodulosus*	Kuril Basin, Sea of Okhotsk, and the adjacent abyssal area of the Pacific Ocean, including the slope of the Kuril-Kamchatka Trench, 3500 m	[21]
22:3(n-5)	2.4%			
22:5(n-5)	18.3%			
23:4(n-7)	0.4%	*Pyrgo* sp.		
24:4(n-8)	0.4%			
24:5(n-9)	1.7%			
26:4(n-6)	0.9%			
26:5(n-3)	0.9%			
Echinodermata				
22:6(n-2)	1.6–0.33%	sea urchin *Eremicaster vicinus* sea star *Kamptosoma abyssale*	Kuril–Kamchatka Trench, 5200 and 6300 m	[22]
22:5(n-5)	2.8%	sea star *Eremicaster* sp.		[50]
24:6(n-3)	20.5%	ophiuran *Ophiopenia vicina*		
26:7(n-3)	14.5%	ophiuran *Amphiophiura ponderosa*		

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
