# Peer review of "Investigation of Deep-Sea Ecosystems Using Marker Fatty Acids: Sources of Essential Polyunsaturated Fatty Acids in Abyssal Megafauna"

_marinedrugs, 2021, doi:10.3390/md20010017_

Round 1

Reviewer 1 Report

The paper provides an interesting account of information on the fatty acid (FA) composition of abyssal organisms, but as a whole it is of quite a fragmentary nature. As such, general conclusions are often drawn from anecdotal specific information, and expressions like “we suppose (L20), it can be assumed (L127), suggested (L187), may be (L214), suggests (L236), can be linked (L274), allows an assumption (L400), another possible source (L463), are possibly (L489), can be (L534)” add little or none strength to the arguments. I do not have any problem on the author proposing new hypothesis from any sort of previous data, but it should be made crystal clear that they are just hypotheses or speculations. Perhaps adding “potential” to the title (……Potential sources of essential polyunsaturated fatty acids…..) could help from the very beginning. On the other hand, it should be acknowledged that the author recognizes the need of additional studies in other parts of the manuscript (L472).

I miss a discussion on the potential contribution of the metabolism of the different groups to their fatty acid composition (endogenous vs exogenous). There is only one tangential reference to this on L203 and L225. In this sense, note that NMID can be products of endogenous synthesis and not only provided by bacteria (L136, L405). A discussion on the possible contribution of the body contents on the final FA profiles would also be interesting. If there is no emptying period prior to the analyses, the effect of the food would be overestimated. This goes in line with the assessments in L110 and L130 if I understand it properly. These aspects seem much more interesting to me than section 2, whose presence, like that of figure 2, I find hardly if any, justified (can be removed in terms of consistency).

I miss references to support the assessments in L197, L451 and L529. Please note also that the last sentence of the paper is of quite obvious nature and inherent of the very definition of trophic markers.

MEFA are lacking from the list of abbreviations.

Note that many names of species mainly in the nematode section, but along the whole manuscript should be in italics. Note also the lack of consistency when mentioning the FA names, especially arachidonic acid (AA) which is indistinctively written in full and in abbreviation, or cis vaccenic acid in L165 and L182.

Author Response

The paper provides an interesting account of information on the fatty acid (FA) composition of abyssal organisms, but as a whole it is of quite a fragmentary nature. As such, general conclusions are often drawn from anecdotal specific information, and expressions like “we suppose (L20), it can be assumed (L127), suggested (L187), may be (L214), suggests (L236), can be linked (L274), allows an assumption (L400), another possible source (L463), are possibly (L489), can be (L534)” add little or none strength to the arguments. I do not have any problem on the author proposing new hypothesis from any sort of previous data, but it should be made crystal clear that they are just hypotheses or speculations. Perhaps adding “potential” to the title (……Potential sources of essential polyunsaturated fatty acids…..) could help from the very beginning. On the other hand, it should be acknowledged that the author recognizes the need of additional studies in other parts of the manuscript (L472).

This manuscript describes a substantially understudied area of research with hard-to-obtain materials for investigation. Most data were collected from different places and different methods were applied. I think, this explains the somewhat fragmentary appearance of this paper. Thus, the shortage and, sometimes, contradictions of experimental data made me use more cautious expressions like “we suppose”, “it can be assumed”, etc.

I miss a discussion on the potential contribution of the metabolism of the different groups to their fatty acid composition (endogenous vs exogenous). There is only one tangential reference to this on L203 and L225. In this sense, note that NMID can be products of endogenous synthesis and not only provided by bacteria (L136, L405). Actually, NMID acids are absent from bacteria, and are synthesized from bacterial 16:1(n-7) and 18:1(n-7) by different invertebrates like Bivalvia. A discussion on the possible contribution of the body contents on the final FA profiles would also be interesting. If there is no emptying period prior to the analyses, the effect of the food would be overestimated. This goes in line with the assessments in L110 and L130 if I understand it properly. These aspects seem much more interesting to me than section 2, whose presence, like that of figure 2, I find hardly if any, justified (can be removed in terms of consistency). Fig. 2 is deleted.

I miss references to support the assessments in L197[20,31 foraminifera], L451[ 14,15,97,98 biomembranes in deep-sea organisms] and L529 it supported by ref.[33-35, as mentioned earlier] Please note also that the last sentence of the paper is of quite obvious nature and inherent of the very definition of trophic markers.

MEFA are lacking from the list of abbreviations: Changed to the correct acronym, FAME.

Note that many names of species mainly in the nematode section, but along the whole manuscript should be in italics. Corrected throughout the text. Note also the lack of consistency when mentioning the FA names, especially arachidonic acid (AA) which is indistinctively written in full and in abbreviation, or cis vaccenic acid in L165 and L182. These names were used as synonyms for better readability.

Reviewer 2 Report

This is a good and well written paper on an important subject - lipid cycle and availability in abyssal zone. It shed some light on the deep ocean life system, which is very important from the ecological point of view. Paper should be published as it is with small editorial changes to be introduced

1./ line 29: the frist sentece should be a part of paragraph's title;

2./ line 189, caption to Table 1: word "Echinodermate" is surplus;

3./ names of bacterial and animal species should be given in italics. This especially considers text between lines 220-229 but also in other parts of the text.

Author Response

This is a good and well written paper on an important subject - lipid cycle and availability in abyssal zone. It shed some light on the deep ocean life system, which is very important from the ecological point of view. Paper should be published as it is with small editorial changes to be introduced

1./ line 29: the first sentence should be a part of paragraph's title: Changed.

2./ line 189, caption to Table 1: word "Echinodermate" is surplus: Changed to Foraminifera and Echinodermata, because both phyla are mentioned in Table.

3./ names of bacterial and animal species should be given in italics. This especially considers text between lines 220-229 but also in other parts of the text: Corrected throughout the text

Round 2

Reviewer 1 Report

Please note that there are scientific names in Table 1 that are not written in italics. I would suggest to review again the text to fix other minor detils like the space in 20:2 Δ5,13, L393. The query about Section 2 has not been rebutted.

Author Response

Please note that there are scientific names in Table 1 that are not written in italics. I would suggest to review again the text to fix other minor details like the space in 20:2 Δ5,13, L393. The query about Section 2 has not been rebutted.

I corrected text according minor notes. About section 2:

I added 5 Lines to this section and change Header on 2. Some notes on samples used for lipid extraction and the methods of fatty acid analysis

Deep-sea samples are not always available in sufficient amounts for dissection; some of them are subject to damage and loss of integrity during collection and retrieval to the surface. Small animal are difficult or impossible to dissect for separation of tissues, so they are extracted whole. In the cited articles for analysis of holothurian lipids used body wall; for sea stars tube feet; and for sea urchins total guts, after washing from sediments. In mollusks, muscle tissue was analyzed. More detailed information about sample treatment can be found in cited articles.